# Training Federated GANs with Theoretical Guarantees: A Universal Aggregation Approach

## Abstract

Recently, Generative Adversarial Networks (GANs) have demonstrated their potential in federated learning, i.e., learning a centralized model from data privately hosted by multiple sites. A federated GAN jointly trains a centralized generator and multiple private discriminators hosted at different sites. A major theoretical challenge for the federated GAN is the heterogeneity of the local data distributions. Traditional approaches cannot guarantee to learn the target distribution, which is a mixture of the highly different local distributions. This paper tackles this theoretical challenge, and for the first time, provides a provably correct framework for federated GAN. We propose a new approach called Universal Aggregation, which simulates a centralized discriminator via carefully aggregating the mixture of all private discriminators. We prove that a generator trained with this simulated centralized discriminator can learn the desired target distribution. Through synthetic and real datasets, we show that our method can learn the mixture of largely different distributions where existing federated GAN methods fail.

## 1 Introduction

Generative Adversarial Networks (GANs) have attracted much attention due to their ability to generate realistic-looking synthetic data (Goodfellow et al., 2014; Zhang et al., 2018; Liu et al., 2019b; Shaham et al., 2019; Dai et al., 2017; Kumar et al., 2017). In order to obtain a powerful GAN model, one needs to use data with a wide range of characteristics (Qi, 2019). However, these diverse data are often owned by different sources, and to acquire their data is often infeasible. For instance, most hospitals and research institutions are unable to share data with the research community, due to privacy concerns (Annas et al., 2003; Mercuri, 2004; lex, 2014; Gostin et al., 2009) and government regulations (Kerikmäe, 2017; Seddon & Currie, 2013).

To circumvent the barrier of data sharing for GAN training, one may resort to Federated Learning (FL), a promising new decentralized learning paradigm (McMahan et al., 2017). In FL, one trains a centralized model but only exchanges model information with different data sources. Since the central model has no direct access to data at each source, privacy concerns are alleviated (Yang et al., 2019; Kairouz et al., 2019). This opens the opportunity for a *federated GAN*, i.e., a centralized generator with multiple local and privately hosted discriminators (Hardy et al., 2019). Each local discriminator is only trained on its local data and provides feedback to the generator w.r.t. synthesized data (e.g., gradient). A federated GAN empowers GAN with much more diversified data without violating privacy constraints.

Despite the promises, a convincing approach for training a federated GAN remains unknown. The major challenge comes from the non-identical local distributions from multiple data sources/entities. The centralized generator is supposed to learn a mixture of these local distributions from different entities, whereas each discriminator is only trained on local data and learns one of the local distributions. The algorithm and theoretical guarantee of traditional single-discriminator GAN (Goodfellow et al., 2014) do not easily generalize to this federated setting. A federated GAN should integrate feedback from local discriminators in an intelligent way, so that the generator can 'correctly' learn the mixture distribution. Directly averaging feedbacks from local discriminators (Hardy et al., 2019) results in a strong bias toward common patternsowever, such non-identical distribution setting is classical in federated learning (Zhao et al., 2018; Smith et al., 2017; Qu et al., 2020) and characteristic of local data improves the diversity of data.

In this paper, we propose *the first theoretically guaranteed federated GAN*, that can correctly learn the mixture of local distributions. Our method, called Universal Aggregation GAN (UA-GAN), focuses on the odds value rather than the predictions of local discriminators. We simulate an unbiased centralized discriminator whose odds value approximates that of the mixture of local discriminators. We prove that by aggregating gradients from local discriminators based on the odds value of the central discriminator, we are guaranteed to learn the desired mixture of local distributions.

A second theoretical contribution of this paper is an analysis of the quality of the federated GAN when the local discriminators cannot perfectly learn with local datasets. This is a real concern in a federated learning setting; the quantity and quality of local data can be highly variant considering the limitation of real-world institutions/sites. Classical theoretical analysis of GAN (Goodfellow et al., 2014) assumes an optimal discriminator. To understand the consequence of suboptimal discriminators, we develop a novel analysis framework of the Jensen-Shannon Divergence loss (Goodfellow et al., 2014; Lin, 1991) through the odds value of the local discriminators. We show that when the local discriminators behave suboptimally, the approximation error of the learned generator deteriorates linearly to the error.

It is worth noting that our theoretical result on suboptimality also applies to the classical GAN. To the best of our knowledge, *this is the first suboptimality bound on the federated or classical GAN*. In summary, the contributions are threefold.

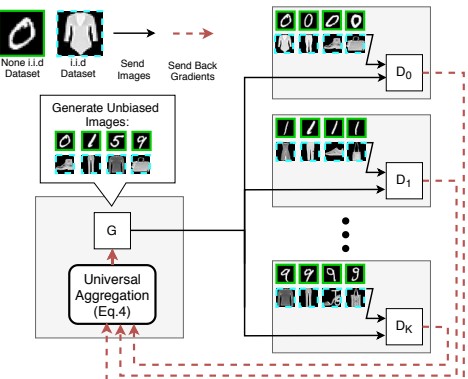

- We propose UA-GAN, a novel federated GAN approach that aggregates feedback from local discriminators through their odds value rather than posterior probability.
- We prove that UA-GAN correctly learns the mixture of local distributions when they are perfectly modeled by local discriminators.
- We prove when the discriminators are suboptimal in modeling their local distributions, the generator's approximation error is also linear. We also show that our bound is tight.

We show with various experiments that our method (UA-GAN) outperforms the state-of-the-art federated GAN approaches both qualitatively and quantitatively.

Training on large scale heterogeneous datasets makes it possible to unleash the power of GANs. Federated GANs show their promise in utilizing unlimited amount of sensitive data without privacy and regulatory concerns. Our method, as the first theoretically guaranteed GAN, will be one step further in building such a foundation. Fig. 1 shows the workflow of UA-GAN.

Figure 1: **UA-GAN framework**: The data from multiple entities may share some common distribution while retain its own individual distribution. Here, we use the MNIST dataset that the 10 digits belong to different data center as common pattern, and MNIST Fashion dataset that randomly split into different data centers as distinct features. (see Sec. 4.2 for details). We can see the UA-GAN carefully aggregates the feedback from different entities and manage to digest the universal distribution generate unbiased synthetic data.

## 2 RELATED WORK

**The Generative Adversarial Networks (GANs)** have enjoyed much success in various machine learning and computer vision tasks (Zhang et al., 2018; Liu et al., 2019b; Shaham et al., 2019; Dai et al., 2017; Kumar et al., 2017). Numerous methods are proposed for GAN training, such as Spectral Normalization (SN) (Miyato et al., 2018), zero-centered gradient penalty (Mescheder et al., 2018; Thanh-Tung et al., 2019), WGAN (Arjovsky et al., 2017) , WGAN-GP (Gulrajani et al., 2017), WGAN-TS (Liu et al., 2018), WGAN-QC (Liu et al., 2019a) etc. A common approach in practice is the conditional GAN (cGAN) (Mirza & Osindero, 2014), which uses supervision from data (e.g., class labels) to improve GAN's performance.

**Multi-discriminator/-generator GANs** have been proposed for various learning tasks. To train these GANs, one common strategy is to directly exchange generator/discriminator model parameters during training (Xin et al., 2020; Hardy et al., 2019). This is very expensive in communication; a simple ResNet18 (He et al., 2016a) has 11 million parameters (40MB). Closest to us is MD-GAN (Hardy

---

**Algorithm 1** Training Algorithm of UA-GAN.

---

1: **Input:** Batch size $m$, datasets $\{\mathcal{D}_j\}$, size of datasets $\{\pi_j = \frac{n_j}{n}\}$.
2: **Output:** $G, D_j, \forall j \in [K]$.
3: **for** $t = 1, \cdots, T$ **do**
4:     {Work at the central server.}
5:     $G$ generates synthetic data: $\hat{x}_i = G(z_i)$, $i = 1, \cdots, m$.
6:     Send batch of synthetic data $\mathcal{D}_{syn} = \{\hat{x}_1, \cdots, \hat{x}_m\}$ to all $K$ sites.
7:     **for** $j = 1, \cdots, K$ **do**
8:         {Work at each local site.}
9:         Update the local discriminator, $D_j$, using real samples from $\mathcal{D}_j$ and synthetic data batch, $\mathcal{D}_{syn}$, based on Eq. 2.
10:         Output predictions and gradients for synthetic data $D_j(\hat{x}_i)$, $\partial D_j(\hat{x}_i)/\partial \hat{x}_i$, $i = 1, \cdots, m$. Send them to the central server.
11:     **end for**
12:     {Work at the central server.}
13:     Simulate value of $D_{ua}(\hat{x}_i)$ via Eq. 4, $\forall i$.
14:     Update $G$ based on Eq. 5, using gradients from $D_j$'s.
15: **end for**

---

et al., 2019), which aggregates feedbacks (gradients) from local discriminators through averaging. It also swaps parameters between discriminators. None of these methods provide theoretical guarantee as ours. Meanwhile, our method is the only one without model swapping, and thus is much more efficient in bandwidth consumption.

**Federated Learning (FL)** (Kairouz et al., 2019; McMahan et al., 2016) offers the opportunity to integrate sensitive datasets from multiple sources through distributed training. Many works have been done tackling practical concerns in FL, such as convergence under Non-IID data assumption (Yu et al., 2019; Lian et al., 2017; Li et al., 2020), decentralized SGD without freezing parameters (Recht et al., 2011; Nguyen et al., 2018), communication efficiency (Konečnỳ et al., 2016; Li et al., 2019), provable privacy guarantees (Alistarh et al., 2017; Wei et al., 2020). Federated GAN is also of great interest from a federated learning perspective. A successful federated GAN makes it possible to train a centralized model (e.g., a classifier) using data synthesized by the centralized generator. This becomes a solution when existing trained FL model needs to be replaced and updated by advanced machine learning approaches , as one can retrain the model at any time using the generator. It also alleviates some privacy concerns of FL, e.g., the gradient leakage problem (Zhu et al., 2019).

## 3 METHOD

To introduce our algorithm, we first introduce notations and formalize the mixture distribution learning problem. Next, we present our Universal Aggregation approach and prove that it is guaranteed to learn the target mixture distribution. We also analyze the suboptimality of the model when local discriminators are suboptimal. For ease of exposition, we mostly use ordinary GAN to illustrate the algorithm and prove its theoretical properties. At the end of this section, we extend the algorithm, as well as its theoretical guarantees, to conditional GAN (cGAN) Mirza & Osindero (2014). The empirical results in this work are established on the cGAN since its training is much more controllable, thanks to the additional supervision by the auxiliary variable (e.g., classes of images).

**Notations and problem formulation.** We assume a cross-silo FL setting Kairouz et al. (2019), i.e., $K$ entities hosting $K$ private datasets $\mathcal{D}_1, ..., \mathcal{D}_K$, with size $n_1, \cdots, n_K$. The total data size $n = \sum_{j=1}^{K} n_j$. The overall goal is to learn a target mixture distribution

$$p(x) = \sum\nolimits_{j=1}^{K} \pi_j p_j(x),\tag{1}$$

in which a component distributions $p_j(x)$ is approximated by the empirical distribution from the $j$-th local dataset $\mathcal{D}_j$. The mixing weight $\pi_j$ is computed using the fraction of dataset $\mathcal{D}_j$: $\pi_j = n_j/n$. In general, different mixture components $p_i(x)$ and $p_j(x)$ may (but not necessarily) be non-identical, namely, $\exists x$, such that $p_i(x) \neq p_j(x)$.

**Universal Aggregation GAN:** Now we are ready to introduce our multi-discriminator aggregation framework. A pseudo-code of UA framework can be found in Algorithm 1. We have a centralized

(conditional) generator $G(z)$ seeking to learn the global distribution $p(x)$. In each local site, a discriminator $D_j(x)$ has access to local dataset $\mathcal{D}_j$. Note data in $\mathcal{D}_j$ are only sampled from $p_j(x)$. During training, the generator generates a batch of synthetic data to all $K$ sites. The $j$-th discriminator seeks to minimize the cross entropy loss of GAN from a local perspective Goodfellow et al. (2014):

$$\max_{D_j} V(G, D_j) = \mathbb{E}_{x \sim p_j(x)}[\log D_j(x)] + \mathbb{E}_{z \sim \mathcal{N}(0, I_d)}[\log(1 - D_j(G(z)))] \tag{2}$$

To formalize the generator training, we first introduce odds value. It is an essential quantity for our algorithm and its analysis.

**Definition 1** (odds value). *Given a probability $\phi \in (0, 1)$, its odds value is $\Phi(\phi) \triangleq \frac{\phi}{1-\phi}$. Note the definition requires $\phi \neq 1$.*

Also it is straightforward to see $\phi = \frac{\Phi(\phi)}{1 + \Phi(\phi)}$.

The central idea of UA Framework is to simulate a centralized discriminator $D_{ua}(x)$ which behaves like the mixture of all local discriminators (in terms of odds value). A well behaved $D_{ua}(x)$ can then train the centralized generator $G$ using its gradient, just like in a classical GAN.

We design $D_{ua}$ so that its odds value $\Phi(D_{ua}(x))$ is identical to the mixture of the odds values of local discriminators:

$$\Phi(D_{ua}(x)) = \sum_{j=1}^{K} \pi_j \Phi(D_j(x)). \tag{3}$$

Given $\Phi(D_{ua}(x))$, we can compute $D_{ua}(x)$ as

$$D_{ua}(x) = \frac{\Phi(D_{ua}(x))}{1 + \Phi(D_{ua}(x))} \tag{4}$$

Once the central discriminator $D_{ua}$ is simulated, the generator can be computed by minimizing the generator loss :

$$\min_{G} V(G, D_{ua}) = \mathbb{E}_{x \sim p(x)}[\log D_{ua}(x)] + \mathbb{E}_{z \sim \mathcal{N}(0, I_d)}[\log(1 - D_{ua}(G(z)))] \tag{5}$$

Note that mathematically, Eq. (5) can be directly written in terms of local discriminators $D_j$'s (by substituting in Eqs (3) and (4)). In implementation, the simulated central discriminator can be written as a pytorch or tensorflow layer.

**Intuition.** The reason we define $D_{ua}$'s behavior using a mixture of odds values instead of a mixture of the predictions is mathematical. It has been shown in Goodfellow et al. (2014) that a perfect discriminator learning a data distribution $p(x)$ and a fixed generator distribution $q(x)$ satisfies $D(x) = \frac{p(x)}{p(x)+q(x)}$. It can be shown that only with the odds value equivalency, this optimal solution of the central discriminator $D(x)$ can be recovered if each individual discriminator is optimal, i.e., $D_j(x) = \frac{p_j(x)}{p_j(x)+q(x)}$. This is not true if we define the central discriminator behavior using the average prediction, i.e., $D_{ua} = \sum_j \pi_j D_j$. More details can be found in Theorem (4) and its proof.

**Remark 1** (Privacy Safety). *For federated learning, it is essential to ensure information of real data are not leaked outside the local site. This privacy safety is guaranteed in our method. To optimize $G$ w.r.t. Eq. (5), we only need to optimize the second term and use gradient on synthetic images $G(z)$ from local discriminators.*

One important concern is about the optimal discriminator condition. $D_{ua}(x)$ is designed to be optimal only when $D_j$'s are optimal. We need to investigate the consequence if the local discriminators $D_j$'s are suboptimal. We will provide an error bound of the learned distribution w.r.t., the suboptimality of $D_j$'s in Corollary (2).

### 3.1 THEORETICAL ANALYSIS OF UA-GAN

In this section, we prove the theoretical guarantees of UA-GAN. First, we prove the correctness of the algorithm. We show that if all local discriminators can perfectly learn from their local data, the algorithm is guaranteed to recover the target distribution (Eq. (1)). Second, we discuss the quality of the learned generator distribution when the local discriminators are suboptimal, due to real-world

constraints, e.g., insufficient local data. We show that the error of the learned distribution is linear to the suboptimality of the local discriminators. All results in this section are for original (unconditional) GANs. But they can be extended to conditional GANs (see Sec. (3.2)). Due to space constraints, we only state the major theorems and leave their proofs to the supplemental material.

CORRECTNESS OF UA-GAN

The correctness theorem assumes all local discriminators behave optimally.

**Assumption 1** (Optimal Local Discriminator). *Assume all local discriminators are* optimal, *i.e., they learn to predict whether a data is true/fake perfectly. Let $q(x)$ be the probability of the current generator $G$. A local discriminator is optimal iff $D_j(x) = \frac{p_j(x)}{q(x)+p_j(x)}$.*

Theorem (4) states the unbiasedness of UA-GAN: with optimal local discriminators, the generator learns the target distribution.

**Theorem 1** (Correctness). *Suppose all discriminators $D_j$'s are optimal. $D_{ua}(x)$ is computed via Eq. (3). Denote by $q$ the density function of data generated by $G$. Let $q^*(\cdot)$ be the optimal distribution w.r.t. the Jenson Shannon divergence loss :*

$$q^* := \arg\min_q L(q) = \mathbb{E}_{x \sim p(x)}[\log D_{ua}(x)] + \mathbb{E}_{x \sim q(x)}[\log(1 - D_{ua}(x)]. \qquad (6)$$

*We have $q^*$ equals to the true distribution, formally, $q^* = p$.*

The proof mainly establishes that when $D_j$'s are optimal, $D_{ua}$ is also optimal. With the optimality of $D_{ua}$ proved, the rest follows the correctness proof of the classic GAN (Theorem 1 in Goodfellow et al. (2014)). More details are in the supplemental material.

ANALYSIS OF THE SUBOPTIMAL SETTING

In centralized learning setting, an optimal discriminator is a reasonable assumption since the model has access to all (hopefully sufficient) data. However, in federated GAN setting, available data in some site $\mathcal{D}_j$ may be limited. One natural question in this limited data scenario is: how would the generator behave if some local discriminators are suboptimal? We address this theoretical question.

We first focus on a single discriminator case. We show the behavior of a perturbed version of Jensen-Shannon divergence loss Guha et al. (2007); Lin (1991); Csiszár et al. (2004). The suboptimality of a central discriminator $D(x)$ is measured by the deviation in terms of the odds value. Denote by $q(x)$ the generator distribution of the current $G$. Ideally, the odds value of an optimal discriminator should be $p(x)/q(x)$. We show that a suboptimal $D$ with $\delta$ deviation from the ideal odds value will result in $O(\delta)$ suboptimality in the target distribution.

**Theorem 2** (Suboptimality Bound for a Single Discriminator). *Suppose a discriminator $\widetilde{D}(x)$ is a perturbed version of the optimal discriminator $D(x)$, s.t. $\Phi(\widetilde{D}(x)) = \Phi(D(x))\xi(x)$ with $|1 - \xi(x)| \leq \delta$ and $\delta \leq 1/8$. Let $q^*$ be the optimal distribution of the Jensen-Shannon divergence loss based on the perturbed discriminator*

$$q^* := \arg\min_q L(q) = \mathbb{E}_{x \sim p(x)}[\log \widetilde{D}(x)] + \mathbb{E}_{x \sim q(x)}[\log(1 - \widetilde{D}(x)]. \qquad (7)$$

*Then $q^*$ satisfies $|q^*(x)/p(x) - 1| \leq 16\delta, \forall x.$*

This theorem shows that the ratio of the learned distribution $q^*$ is close to the target true distribution $p$ when the suboptimality of $D_j$ is small. To the best of our knowledge, this is the first bound on the consistency of Jensen-Shannon divergence with suboptimal discriminator, even for a classical GAN.

Next, we show that the bound is also tight.

**Theorem 3** (Tightness of the Bound in Theorem (5)). *Given a perturbed discriminator $\widetilde{D}(x)$ of the optimal one $D(x)$, s.t. $\Phi(\widetilde{D}(x)) = \Phi(D(x))\xi(x)$ with $|\xi(x) - 1| \geq \gamma$ and $\gamma \leq 1/8$. The optimal distribution $q^*$ as in Eq. (12) satisfies $|q^*(x)/p(x) - 1| \geq \gamma/16, \forall x.$*

Next we extend these bounds for a single discriminator to our multi-discriminator setting. This is based on Theorem 5 and the linear relationship between the local discriminators and the central discriminator.

**Corollary 1** (Suboptimality Bound for UA-GAN). *Assume suboptimal local discriminators $\widetilde{D_j}(x)$ are the perturbed versions of the optimal ones $D_j(x)$. And the suboptimality is bounded as: $\Phi(\widetilde{D_j}(x)) = \Phi(D_j(x))\xi_j(x)$ with $|\xi_j(x) - 1| \leq \delta \leq 1/8, \forall x$. The centralized discriminator $\widetilde{D_{ua}}(x)$ is computed using these perturbed local discriminators such that $\Phi(\widetilde{D_{ua}}(x)) = \sum_{j=1}^{K} \pi_j \Phi(\widetilde{D_j}(x))$. Let $q^*$ be the optimal distribution of the Jensen-Shannon divergence loss based on the perturbed UA discriminator $\widetilde{D_{ua}}$*

$$q^* := \arg\min_q L(q) = \mathbb{E}_{x \sim p(x)}[\log \widetilde{D_{ua}}(x)] + \mathbb{E}_{x \sim q(x)}[\log(1 - \widetilde{D_{ua}}(x)]. \tag{8}$$

*Then $q^*$ satisfies $|q^*(x)/p(x) - 1| = O(\delta)$. In particular, the optimal distribution $q^*(x)$ has $O(\delta)$ total variation distance to the target distribution $p(x)$.*

Note that the lowerbound of the suboptimality for single discriminator (Theorem 6)) can also be extended to UA-GAN similarly.

**Remark 2.** *The consistency gap in Corollary (2) assumes a uniform suboptimality bounded for all local discriminators. In practice, such assumption may not be informative if the sizes of $\mathcal{D}_j$'s data are highly imbalanced. It would be interesting to relax such assumption and investigate the guarantees of UA-GAN w.r.t. the expected suboptimality of $D_j$'s.*

### 3.2 Universal Aggregation Framework for Conditional GAN

Our algorithm and analysis on unconditional GANs can be generalized to the more practical Conditional GAN Mirza & Osindero (2014). A conditional GAN learns the joint distribution of $p(x, y)$. Here $x$ represents an image or a vectorized data, and $y$ is an auxiliary variable to control the mode of generated data (e.g., the class label of an image/data). Conditional GAN is much better to train in practice and is the common choice in most existing works. This is indeed the setting we use in our experiments.

The target distribution of the learning problem becomes a joint mixture distribution:

$$p(x, y) = \sum_j \pi_j \omega_j(y) p_j(x, y),$$

in which $\pi_j = n_j/n$ and $\omega_j(y)$ is the proportion of class $y$ data within the $j$-th local dataset $\mathcal{D}_j$. We assume $\pi_j$, and the dictionary of $y$ and its fractions in each $\mathcal{D}_j$, $\omega_j(y)$ are known to the public. In practice, such knowledge will be used for generating $y$. Formally, $y \sim \sum_{j=1}^{K} \pi_j \omega_j(y)$.

To design the UA-GAN for the conditional GAN, the odds value aggregation in formula Eq. (3) needs to be adapted to:

$$\Phi(D_{ua}(x|y)) = \sum_{j=1}^{K} \pi_j \omega_j(y) \Phi(D_j(x|y)).$$

The computation of $D_{ua}(x|y)$ and the update of $G$ and $D_j$'s need to be adjusted accordingly. The theoretical guarantees for unconditional GANs can also be established for a conditional GAN. Due to space limitation, we leave details to the supplemental material.

## 4 Experiments

On synthetic and real-world datasets, we verify that UA-GAN can learn the target distribution from both i.i.d and non-identical local datasets. We focus on conditional GAN Mirza & Osindero (2014) setting as it is the common choice in practice.

### 4.1 Synthetic Experiment

We evaluate UA-GAN on a toy dataset. See Fig. 2 first row. The toy dataset has 4 datasets, generated by 4 Gaussians centered at (10, 10), (10,-10), (-10,10), (-10,-10) with variance 0.5. Data samples are shown as blue points. The central generator takes Gaussian noise centered at $(0, 0)$ with variance of 0.5 (green points) and learns to transform them into points matching the mixture distribution (orange points). The first figure shows the generator successfully recovers the Gaussian mixture. The contours

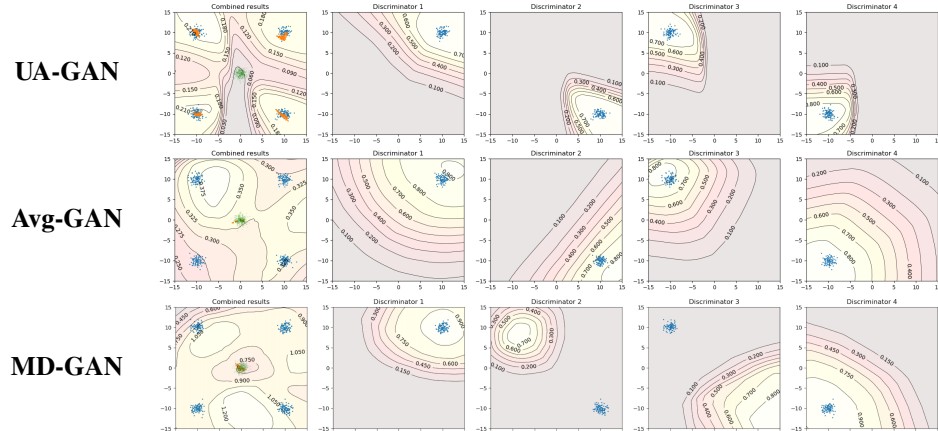

Figure 2: Results on a toy dataset by UA-GAN, Avg-GAN and MD-GAN. UA-GAN can learn four Gaussians, whereas Avg-GAN and MD-GAN fail.

| Dataset | non-identical Mnist + Fashion | | | non-identical Mnist + Font | | |
|---|---|---|---|---|---|---|
| | Accuracy↑ | IS↑ | FID↓ | Accuracy↑ | IS↑ | FID↓ |
| Real | 0.943 | $3.620 \pm 0.021$ | 0 | 0.994 | $2.323 \pm 0.011$ | 0 |
| Centralized GAN | 0.904 | $3.437 \pm 0.021$ | 8.35 | 0.979 | $1.978 \pm 0.009$ | 17.62 |
| Avg GAN | 0.421 | $4.237 \pm 0.023$ | 72.80 | 0.822 | $1.517 \pm 0.004$ | 85.81 |
| MD-GAN | 0.349 | $2.883 \pm 0.020$ | 102.00 | 0.480 | $\mathbf{2.090 \pm 0.007}$ | 69.63 |
| UA-GAN | **0.883** | $\mathbf{3.606 \pm 0.020}$ | **24.60** | **0.963** | $1.839 \pm 0.006$ | **24.73** |

Table 1: Quantitative results on non-identical mixture datasets. UA-GAN achieves better result compared with the Avg GAN and MD-GAN's aggregation method.

show the central discriminator $D_{ua}$ calculated according to Eq. 3. The generated points (orange) are evenly distributed near the 4 local Gaussians' centers. The 2nd to 5th figures show the prediction of the local discriminators ($D_j$'s), overlaid with their respective samples $\mathcal{D}_j$ in blue.

Meanwhile, we show in the second row the results of the average scheme, called Avg-GAN, which averages local discriminators' outputs to train the generator. The first figure shows that the learned distribution $D_{avg} = \frac{1}{K} \sum_j D_j$ is almost flat. The generated samples (orange) collapse near the origin. Although each individual discriminator can provide valid gradient information (see 2nd to 5th figures for $D_j$'s), naively averaging their outputs cannot achieve the target distribution, when the distributions are symmetric. We show the results of the MD-GAN by Hardy et al. (2019) in the third row. MD-GAN also adopts the average scheme, but randomly shuffle discriminators parameters during training. Similar to Avg-GAN, MD-GAN cannot learn the four Gaussians.

## 4.2 REAL-WORLD MIXTURE DATASETS

We evaluate our method on several mixture datasets, both i.i.d and non-identical.

**Datasets.** Three real-world datasets are utilized to construct the mixture datasets: MNIST LeCun et al. (1998), Fashion-MNIST Xiao et al. (2017), and Font dataset. We create the Font dataset from 2500+ fonts of digits taken from the Google Fonts database Mo (2002). Similar to MNIST, it consists of 10 classes of $28 \times 28$ grayscale images, with 60k samples for training and 29k samples for test. To make the differences more clear between font and handwrite images, we highlight the Font images with a circle when build the dataset. Using these foundation datasets, we create 2 different mixture datasets with non-identical local datasets: (1) non-identical MNIST+Fashion; (2) non-identical MNIST+Font. We uniformly sample Fashion/Font data for all 10 distributed sites. These are common patterns across all sites. Meanwhile, for each individual site, we add MNIST data with a distinct class among $0 \sim 9$. These data are distinguishable features for different sites. Ideally, a federated GAN should be able to

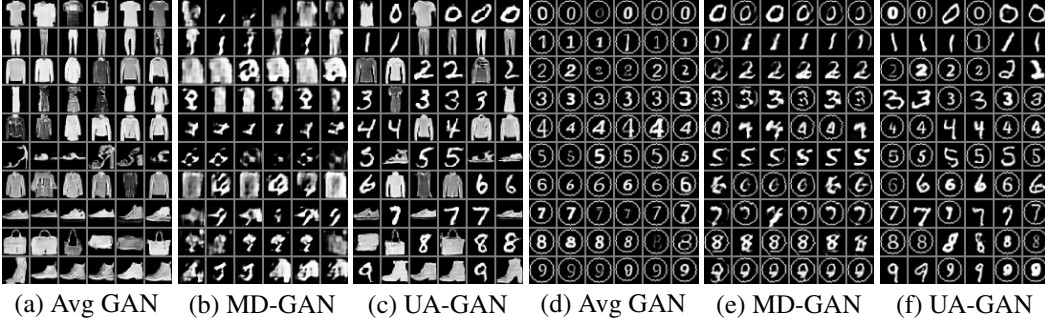

|  (a) Avg GAN |  (b) MD-GAN |  (c) UA-GAN |  (d) Avg GAN |  (e) MD-GAN |  (f) UA-GAN |

Figure 3: Synthetic images on the non-identical MNIST+Fashion ((a),(b),(c)) and MNIST+font datasets ((d),(e),(f)).

learn both the common patterns and the site-specific data. Please see supplemental material for more details and samples from these mixture datasets.

**Baselines.** We compare UA-GAN with Avg-GAN and another SOTA federated GAN, MD-GAN Hardy et al. (2019). We also include two additional baselines: **Centralized GAN** trains using the classic setting, i.e., one centralized discriminator that has access to all local datasets. Comparing with this baseline shows how much information we lose in the distributed training setting. Another baseline is **Real**, which essentially uses real data from the mixture datasets for evaluation. This is the upper bound of all GAN methods. More details can be found in supplementary material.

**Evaluation metrics.** We adopt **Inception score (IS)** Salimans et al. (2016), **Frechet Inception distance (FID)** Heusel et al. (2017) to measure the quality of the generated images. As GAN is often used for training downstream classifiers, we also evaluate the methods by training a classifier using the generated data and report the **Classification Accuracy** as an evaluation metric. This is indeed very useful in federated learning; a centralized classifier can be trained using data generated by federated GANs without seeing the real data from private sites.

**Discussion.** The quantitative results on the two non-identical mixture datasets are shown in Table 1. UA-GAN significantly outperforms the other two federated GANs, Avg-GAN and MD-GAN. Its performance is even close to the centralized GAN, showing that our algorithm manages to mitigate the challenge of distributed training to a satisfying degree.

The superior performance of UA-GAN can be illustrated by qualitative examples in Fig. 3. On MNIST+Fashion dataset (subfigures a-c), the average aggregation strategy used by Avg-GAN could not effectively aggregate outputs of 'non-identical' local discriminators. Therefore, it only learns to generate the common patterns, e.g., Fashion images (Fig. 3(a)). MD-GAN fails to produce high quality images (Fig. 3(b)), probably because the discriminator switch makes the training not stable enough. Meanwhile, our UA-GAN is able to generate the mixture with both common patterns (Fashion images) and site-specific images (different digits from MNIST) with high quality. Similar phenomenon can be observed for MNIST+Font (subfigures d-f). Avg-GAN only learns the common pattern (computerized fonts from Font dataset), MD-GAN gives low quality images whereas UA-GAN can also learns the high-quality site-specific handwriting digits (MNIST).

Note that we also compare the methods on mixture datasets with i.i.d local distributions, i.e., all local datasets are sampled in a same way from the real datasets. In an i.i.d setting, all federated GANs and the centralized GAN perform similarly. More results will be included in the supplemental material.

## 5  CONCLUSION AND FUTURE WORK

In this work, we proposed a provably correct federated GAN. It simulates a centralized discriminator via carefully aggregating the feedbacks of all local discriminators. We proved that the generator learns the target distribution. We also analyzed the error bound when the discriminator is suboptimal due to local dataset limitation. A well-trained federated GAN enpowers GANs to learn from diversified datasets. It can also be used as a data provider for training task-specific models without accessing or storing privacy sensitive data.

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

In this supplemental material, we provide proofs of the theorems in the main paper (Sec. A). We also provide additional experimental details and results (Sec. B).

## A  PROOFS OF THEOREMS IN SECTION 3

We recall the definition of odds value.

**Definition 2** (odds value)**.** *Given a probability $\phi \in (0, 1)$, its odds value is $\Phi(\phi) \triangleq \frac{\phi}{1-\phi}$. Note the definition requires $\phi \neq 1$.*

## A.1 ANALYSIS OF OPTIMAL DISCRIMINATOR

We recall the theorem of the correctness of UA-GAN.

**Theorem 4** (Correctness). *Suppose all discriminators $D_j$'s are optimal. $D_{ua}(x)$ is computed via Eq. (11). The optimal distribution of the Jenson-Shannon divergence loss:*

$$\arg\min_q L(q) = \mathbb{E}_{x \sim p(x)}[\log D_{ua}(x)] + \mathbb{E}_{x \sim q(x)}[\log(1 - D_{ua}(x)] \tag{9}$$

*is $q^* = p$ where $q$ is the density (mass) function of $G(z)$.*

To prove the theorem, we first introduce the following Lemma, which is similar to Proposition 1 in Goodfellow et al. (2014). We include the Lemma and Proof here for completeness.

**Lemma 1.** *When generator $G$ is fixed, the optimal discriminator $D_j(x)$ is :*

$$D_j(x) = \frac{p_j(x)}{p_j(x) + q(x)} \tag{10}$$

**Proof**:

$$\max_{D_j} V_j(D_j) = \max_{D_j} \int_x p_j(x) log D_j(x) + q(x) log(1 - D_j(x)) dx$$

$$\leq \int_x \max_{D_j} \{p_j(x) log D_j(x) + q(x) log(1 - D_j(x))\} dx$$

by setting $D_j(x) = \frac{p_j(x)}{p_j(x) + q(x)}$ we can maximize each component in the integral thus make the inequality hold with equality. $\qquad\square$

**Proof of Theorem 4**: Suppose in each training step the discriminator achieves its maximal criterion in Lemma 1, the simulated $D_{ua}(x)$ becomes:

$$D_{ua}(x) = \frac{\sum_j^K \pi_j(y) \frac{D_j^*(x)}{1 - D_j^*(x)}}{1 + \sum_j^K \pi_j(y) \frac{D_j^*(x)}{1 - D_j^*(x)}} \tag{11}$$

Given that discriminators $D_1, ..., D_K$ behave optimally, the value of $\frac{D_j(x)}{1 - D_j(x)} = \frac{p_j(x)}{q(x)}$ which implies $\sum_j \frac{\pi_j D_j(x)}{1 - D_j(x)} = \frac{\sum_j \pi_j p_j(x)}{q(x)} = \frac{D_{ua}(x)}{1 - D_{ua}(x)}$. By the aggregation formula 11, the simulated discriminator will be $D_{ua}(x) = \frac{\sum_j \pi_j p_j(x)}{\sum_j \pi_j p_j(x) + q(x)} = \frac{p(x)}{p(x) + q(x)}$. Suppose in each training step the discriminator achieves its maximal criterion in Lemma 1, the loss function for the generator becomes:

$$\min_a L(q) = \mathbb{E}_{x \sim p(x)}[\log D(x)] + \mathbb{E}_{\hat{x} \sim q(\hat{x}|y)}[\log(1 - D(\hat{x})]$$

$$= \mathbb{E}_{x \sim p(x}[\log D(x)] + \mathbb{E}_{\hat{x} \sim q(\hat{x})}[\log(1 - D(\hat{x}))]$$

$$= \int_x p(x) \log \frac{p(x)}{p(x) + q(x)} + q(x) \log \frac{q(x)}{p(x) + q(x)} dx$$

The above loss function has optimal solution of $q$ due to Theorem 1 in Goodfellow et al. (2014). $\square$

## A.2 ANALYSIS OF SUB-OPTIMAL DISCRIMINATOR

We provide proofs of Theorems 5, 6, and Corollary 2.

**Theorem 5** (Suboptimality Bound for a Single Discriminator). *Suppose a discriminator $\widetilde{D}(x)$ is a perturbed version of the optimal discriminator $D(x)$, s.t. $\Phi(\widetilde{D}(x)) = \Phi(D(x))\xi(x)$ with $|1 - \xi(x)| \leq \delta$ and $\delta \leq 1/8$. Let $q^*$ be the optimal distribution of the Jensen-Shannon divergence loss based on the perturbed discriminator*

$$q^* = \arg\min_q L(q) = \mathbb{E}_{x \sim p(x)}[\log \widetilde{D}(x)] + \mathbb{E}_{x \sim q(x)}[\log(1 - \widetilde{D}(x)]. \tag{12}$$

*Then $q^*$ satisfies $|q^*(x)/p(x) - 1| \leq 16\delta, \forall x.$*

**Lemma 2.** *Suppose* $0 < |a|, |b| \leq \frac{1}{8}$, $\log \rho = \log(\frac{1}{2}+a)+b$ *with* $0 < \rho < 1$, *we have* $1-2|a|-2|b| \leq \frac{\rho}{1-\rho} \leq 1+4|a|+4|b|$.

**Proof**:
In the proof we will use following fact:

$$1 + x \leq e^x \leq 1 + 2x, \text{ for } 0 < x < \frac{1}{8} \tag{13}$$

By equation $\log \rho = \log(\frac{1}{2} + a) + b$ we have $\rho = (\frac{1}{2} + a)e^b$ thus:

$$
\begin{aligned}
(\frac{1}{2} - |a|)(1 - 2|b|) &\leq \rho \leq (\frac{1}{2} + |a|)(1 + 2|b|) \\
\frac{1}{2} - |a| - |b| &\leq \rho \leq \frac{1}{2} + |a| + |b| + 2|ab| \\
\frac{1}{2} - |a| - |b| - 2|ab| &\leq 1 - \rho \leq \frac{1}{2} + |a| + |b| \\
\frac{\frac{1}{2} - |a| - |b|}{\frac{1}{2} + |a| + |b|} &\leq \frac{\rho}{1 - \rho} \leq \frac{\frac{1}{2} + |a| + |b| + 2|ab|}{\frac{1}{2} - |a| - |b| - 2|ab|} \\
(1 - 2|a| - 2|b|) &\leq \frac{\rho}{1 - \rho} \leq (1 + 4|a| + 4|b|)
\end{aligned}
\tag{14}
$$

$\square$

**Lemma 3.** *Suppose* $h(x)/p(x) \geq \frac{1}{2}$, *the following loss function is strongly convex:*

$$L(q) = \int_x p(x) \log \frac{h(x)}{h(x) + q(x)} + q(x) \log \frac{q(x)}{q(x) + h(x)} dx \tag{15}$$

**Proof**:
The first order derivative of $L(q)$ is $\frac{\partial L(q)}{\partial q(x)} = \frac{h(x)-p(x)}{q(x)+h(x)} + \log \frac{q(x)}{q(x)+h(x)}$. The second order derivative is $\frac{\partial^2 L(q)}{\partial q(x)^2} = \frac{q(x)(2h(x)-p(x))+h(x)^2}{(q(x)+h(x))^2 q(x)}$. (There is no non-diagonal elements in the Hessian). $\square$

The proof for Theorem 5 is shown below. The theorem focuses on Jensen-Shannon Divergence loss Guha et al. (2007); Lin (1991). We stress that the analysis is general and works for both general GAN and conditional GAN.

**Proof of Theorem 5**:
Note $\Phi(D(x)) = \frac{p(x)}{q(x)}$. The $\xi(x)$ perturbed odds value gives $\Phi(\widetilde{D}(x)) = \Phi(D(x))\xi(x) = \frac{p(x)\xi(x)}{q(x)}$. Let $h(x) = p(x)\xi(x)$, we have $1 - \delta \leq \frac{h(x)}{p(x)} \leq 1 + \delta$ and $1 - \delta \leq \frac{p(x)}{h(x)} \leq 1 + 2\delta$. The perturbed value of discriminator will be $\widetilde{D}(x) = \frac{h(x)}{h(x)+q(x)}$. The loss function that the generator distribution $q(x)$ seeks to minimize a perturbed Jensen-Shannon Divergence loss:

$$L(q) = \int_x p(x) \log \frac{h(x)}{h(x) + q(x)} + q(x) \log \frac{q(x)}{h(x) + q(x)} dx + \lambda(\int_x q(x)dx - 1). \tag{16}$$

where $\lambda$ represents the Lagrangian Multiplier. In Lemma 3 we verify the convexity of this loss function with regulated behavior of $h(x)$. The derivative of $L(q)$ w.r.t. $q(x)$ is:

$$-\frac{p(x)}{q(x) + h(x)} + \log(q(x)) + 1 - \log(q(x) + h(x)) - \frac{q(x)}{q(x) + h(x)} + \lambda$$

which needs to be 0 for all value of $x$ Boyd & Vandenberghe (2004). Thus we have following equation:

$$\frac{h(x) - p(x)}{q(x) + h(x)} + \log \frac{q(x)}{q(x) + h(x)} = -\lambda \tag{17}$$

Since above equation holds for all $x$. Let $\frac{h(x)-p(x)}{q(x)+h(x)} = \Delta(x)$ which has value bounded by $-\delta \leq \Delta(x) \leq \delta$. we can multiply $p(x) + q(x)$ on both side and integral over $x$:

$$\int_x (h(x) + q(x))\Delta(x) + (h(x) + q(x)) \log \frac{q(x)}{q(x) + h(x)} dx = -\int_x \lambda(h(x) + q(x))dx$$

which gives us $\Delta_1 + \Delta_2 = -2\lambda$ where:

$$\Delta_1 = \frac{1}{2}\int_x (h(x) + q(x))\Delta(x)dx, \quad \Delta_2 = \frac{1}{2}\int_x (h(x) + q(x))\log\frac{q(x)}{q(x) + h(x)}dx$$

Plugging in above derived value of $\lambda$ back into Equation 21 we have:

$$\log\frac{q(x)}{q(x) + p(x)} = \Delta_1 + \Delta_2 - \Delta(x) \tag{18}$$

By the uniform upper bound on $|\Delta(x)| \leq 2\delta$ we have $1 - \frac{3}{2}\delta \leq e^{\Delta_1} \leq 1 + \frac{3}{2}\delta$ and $1 - \frac{3}{2}\delta \leq e^{\Delta} \leq 1 + \frac{3}{2}\delta$. By taking exponential operation on both sides of Equation 22 we have :

$$\frac{q(x)}{q(x) + h(x)} = e^{\Delta_2} * (1 \pm 4\delta)$$

Thus we can bound the value of $e^{\Delta_2}$ by the equation:

$$q(x) = (q(x) + h(x))(e^{\Delta_2} * (1 \pm 4\delta)) \tag{19}$$

. Due to the fact that the equation holds for all value of $x$, by integrating Equation 19 we have $e^{\Delta_2} = \frac{1}{2}(1 \pm 4\delta)$ which is equivalent to $\Delta_2 = \log(\frac{1}{2} \pm 4\delta)$. Thus $\log\frac{q(x)}{q(x)+h(x)} = \log(\frac{1}{2} \pm 4\delta) \pm \Delta$. By Lemma 2, we have $\frac{q(x)}{p(x)} = 1 \pm 16\delta$. $\qquad\square$

**Theorem 6** (Tightness of the Bound in Theorem (5)). *Given a perturbed discriminator $\widetilde{D}(x)$ of the optimal one $D(x)$, s.t. $\Phi(\widetilde{D}(x)) = \Phi(D(x))\xi(x)$ with $|\xi(x) - 1| \geq \gamma$ and $\gamma \leq 1/8$. The optimal distribution $q^*$ as in Eq. (12) satisfies $|q^*(x)/p(x) - 1| \geq \gamma/16, \forall x$.*

**Proof**:
By similar steps to proof in Theorem 3 we have:

$$L(q) = \int_x p(x)\log\frac{h(x)}{h(x) + q(x)} + q(x)\log\frac{q(x)}{h(x) + q(x)}dx + \lambda(\int_x q(x)dx - 1). \tag{20}$$

By setting the derivative of $L(q)$ w.r.t. $q(x)$ be 0 we have :

$$\frac{h(x) - p(x)}{q(x) + h(x)} + \log\frac{q(x)}{q(x) + h(x)} = -\lambda \tag{21}$$

Let $\frac{h(x)-p(x)}{q(x)+h(x)} = \Delta(x)$. By assumption that $|\xi(x) - 1| \geq \gamma$ we have

$$|\Delta(x)| = \frac{|h(x) - p(x)|}{q(x) + h(x)} = \frac{|\xi(x) - 1|}{\frac{h(x)}{p(x)} + \frac{q(x)}{p(x)}}$$

. Due to the fact that $|\xi(x) - 1| \leq \frac{1}{8}$, we have $\frac{q(x)}{p(x)} \leq 2$. Thus we can bound $|\Delta(x)| \geq \frac{\gamma}{8}$.

Next we analyze following equation, we can multiply $p(x) + q(x)$ on both side of (21) and integral over $x$:

$$\int_x (h(x) + q(x))\Delta(x) + (h(x) + q(x))\log\frac{q(x)}{q(x) + h(x)}dx = -\int_x \lambda(h(x) + q(x))dx$$

which gives us $\Delta_1 + \Delta_2 = -2\lambda$ where:

$$\Delta_1 = \frac{1}{2}\int_x (h(x) + q(x))\Delta(x)dx, \quad \Delta_2 = \frac{1}{2}\int_x (h(x) + q(x))\log\frac{q(x)}{q(x) + h(x)}dx$$

Plugging in above derived value of $\lambda$ back into Equation 21 we have:

$$\log\frac{q(x)}{q(x) + p(x)} = \Delta_1 + \Delta_2 - \Delta(x) \tag{22}$$

Next we analyze the term $\Delta_1 - \Delta(x)$.

$$\begin{aligned}
\Delta_1 - \Delta(x) &= \frac{1}{2} \int_x (h(x) + q(x))\Delta(x)dx - \Delta(x) \\
&= \frac{1}{2} \int_x (h(x) + q(x))\frac{h(x) - p(x)}{q(x) + h(x)}dx - \Delta(x) \\
&= \Delta(x)
\end{aligned}$$

Due to the fact that $|\Delta(x)| > \frac{\gamma}{8}$, We can derive that $|\frac{e^{\Delta_2}}{\frac{1}{2}} - 1| > \frac{\gamma}{16}$. By an analysis similar to Theorem 3 we have $|\frac{q(x)}{p(x)} - 1| \geq \frac{\gamma}{64}$. □

**Corollary 2** (Suboptimality Bound for UA-GAN). *Assume suboptimal local discriminators $\widetilde{D_j}(x)$ are the perturbed versions of the optimal ones $D_j(x)$. And the suboptimality is bounded as: $\Phi(\widetilde{D_j}(x)) = \Phi(D_j(x))\xi_j(x)$ with $|\xi_j(x) - 1| \leq \delta \leq 1/8, \forall x$. The centralized discriminator $\widetilde{D_{ua}}(x)$ is computed using these perturbed local discriminators such that $\Phi(\widetilde{D_{ua}}(x)) = \sum_{j=1}^K \pi_j \Phi(\widetilde{D_j}(x))$. Let $q^*$ be the optimal distribution of the Jensen-Shannon divergence loss based on the perturbed UA discriminator $\widetilde{D_{ua}}$*

$$q^* = \arg\min_q L(q) = \mathbb{E}_{x \sim p(x)}[\log \widetilde{D_{ua}}(x)] + \mathbb{E}_{x \sim q(x)}[\log(1 - \widetilde{D_{ua}}(x)]. \tag{23}$$

*Then $q^*$ satisfies $|q^*(x)/p(x) - 1| = O(\delta)$. In particular, the optimal distribution $q^*(x)$ has $O(\delta)$ total variation distance to the target distribution $p(x)$.*

**Proof**:
Let $v_j$'s be odds values of optimal discriminators $D_j(x)$'s: $v_j = \frac{D_j(x)}{1 - D_j(x)}$ and $\widetilde{v_j}$'s be odds values of suboptimal discriminators $\widetilde{D_j}(x)$'s. It suffices to show $|\frac{\sum_j \pi_j v_j}{\sum_j \pi_j \widetilde{v_j}} - 1| \leq \delta$ and apply Theorem 5. □

# B  ADDITIONAL EXPERIMENTAL DETAILS AND RESULTS

**Implementation Details:**  Here we summarize details of the network we use in the experiments. Our UA-GAN has one centralized generator and multiple local discriminators. The generator consists of two fully-connected layers (for input noise and label, respectively), five residual blocks He et al. (2016b) and three upsampling layers. Each discriminator has two convolutional layers (for image and label, respectively), five residual blocks and three average pooling layers. LeakyReLU activation is used in both generator and discriminators. During training, we apply 1 gradient update of the discriminators in each round. Each model is trained with Adam optimizer for 400 epochs with a batch size of 256. The learning rate is initially 0.0002 and linear decays to 0 from epoch 200. The VGG Simonyan & Zisserman (2014) 11-layer model is used for the downstream classification task. We pad the image to $32 \times 32$ and then randomly crop them to $28 \times 28$ with a batch size of 64 as input. The model is trained with SGD optimizer using a learning rate of 0.01 for 150 epochs.

**Dataset Details:**  One of our foundational datasets is the Font dataset. It is created from 2500+ fonts of digits taken from the Google Fonts database. Similar to MNIST, it consists of 10 classes of $28 \times 28$ grayscale images, with 60k samples for training and 29k samples for test.

Based on the MNIST, Fashion-MNIST and Font dataset, we create both i.i.d mixture datasets and non-identical datasets. Details on non-identical datasets have been provided in the main paper. Here we provide details on two i.i.d datasets. (1) i.i.d MNIST+Fashion; (2) i.i.d MNIST+Font. Each of the 10 distributed sites contains 10% of mixture dataset which is uniformly sampled (without replacement) from MNIST and Fashion/Font.

## B.1  EMPIRICAL RESULTS ON I.I.D. DATASETS

The quantitative results on the i.i.d mixture datasets are shown in Table 2. One can see all three distributed GAN methods have comparable performance. It can also be observed from qualitative examples in Fig. 4 that all three methods achieve similar results. This suggests that all three

---

**Algorithm 2** Precise Training Algorithm of UA-GAN.

---

1: **Input:** Batch size $m$, datasets $\{\mathcal{D}_j\}$, size of datasets $\{\pi_j = \frac{n_j}{n}\}$.
2: **Output:** $G, D_j, \forall j \in [K]$.
3: **for** Number of total training iterations **do**
4:    **for** Number of iterations to train discriminator **do**
5:      {Work at the central server.}
6:      $G$ generates synthetic data: $\hat{x}_i = G(z_i)$, $i = 1, \cdots, m$.
7:      Send batch of synthetic data $\mathcal{D}_{syn} = \{\hat{x}_1, \cdots, \hat{x}_m\}$ to all $K$ sites.
8:      **for** $j = 1, \cdots, K$ **do**
9:        {Work at each local site.}
10:        Uniformly randomly choose $m$ real samples $\{x_1^j, \cdots, x_m^j\}$ from $\mathcal{D}_j$:
11:        Update the parameters of local discriminator $D_j$: $\theta_j$ using

$$\nabla_{\theta_j} \frac{1}{m} \sum_{i=1}^{m} \left[ \log(D_j(x_i^j)) + \log(1 - D_j(\hat{x}_i))) \right]$$

12:      **end for**
13:    **end for**
14:    {Work at each local site.}
15:    $G$ generates synthetic data: $\hat{x}_i = G(z_i)$, $i = 1, \cdots, m$.
16:    Send batch of synthetic data $\mathcal{D}_{syn} = \{\hat{x}_1, \cdots, \hat{x}_m\}$ to all $K$ sites.
17:    **for** $j = 1, \cdots, K$ **do**
18:      {Work at each local site.}
19:      Output predictions and gradients for synthetic data $D_j(\hat{x}_i)$, $\partial D_j(\hat{x}_i)/\partial \hat{x}_i$, $i = 1, \cdots, m$. Send them to the central server.
20:    **end for**
21:    {Work at the central server.}
22:    Simulate value of $D_{ua}(\hat{x}_i)$ via Eq. (4), $\forall i$.
23:    Update parameter of $G$: $\theta_G$ by descending its stochastic gradient:

$$\frac{1}{m} \sum_{i=1}^{m} \frac{\partial \log(1 - D_{ua}(\hat{x}_i))}{\partial D_{ua}(\hat{x}_i)} \frac{\partial D_{ua}}{\partial \Phi(D_{ua})} \sum_{j=1}^{K} \left[ \frac{\partial \Phi(D_{ua})}{\partial D_j(\hat{x}_i)} \frac{\partial D_j(\hat{x}_i)}{\partial \hat{x}_i} \right] \frac{\partial \hat{x}_i}{\partial \theta_G}$$

24: **end for**
   The gradient-based updates can use any standard gradient-based learning rule.

---

approaches can be used to train distributed GAN when datasets have i.i.d. distribution e.g., the data is uniformly shuffled before sent to each discriminator. Note that with a similar performance, the UA-GAN has much smaller communication cost compared to MD-GAN since the UA-GAN does not swap model parameters during training process.

| Dataset | i.i.d Mnist + Fashion | | | i.i.d Mnist + Font | | |
|---|---|---|---|---|---|---|
| | Accuracy↑ | IS↑ | FID↓ | Accuracy↑ | IS↑ | FID↓ |
| Real | 0.943 | $3.620 \pm 0.021$ | 0 | 0.994 | $2.323 \pm 0.011$ | 0 |
| Centralized GAN | 0.904 | $3.437 \pm 0.021$ | 8.35 | 0.979 | $1.978 \pm 0.009$ | 17.62 |
| Avg GAN | 0.905 | $3.371 \pm 0.026$ | 12.83 | 0.967 | $1.923 \pm 0.006$ | 19.31 |
| MD-GAN | 0.884 | $3.364 \pm 0.024$ | 13.63 | 0.971 | $1.938 \pm 0.008$ | 19.65 |
| **UA-GAN** | 0.908 | $3.462 \pm 0.024$ | 11.82 | 0.970 | $1.934 \pm 0.008$ | 19.18 |

Table 2: The classification accuracy and IS, FID scores on two i.i.d mixture datasets. All of the three architecture could learn the right distribution with i.i.d datasets.

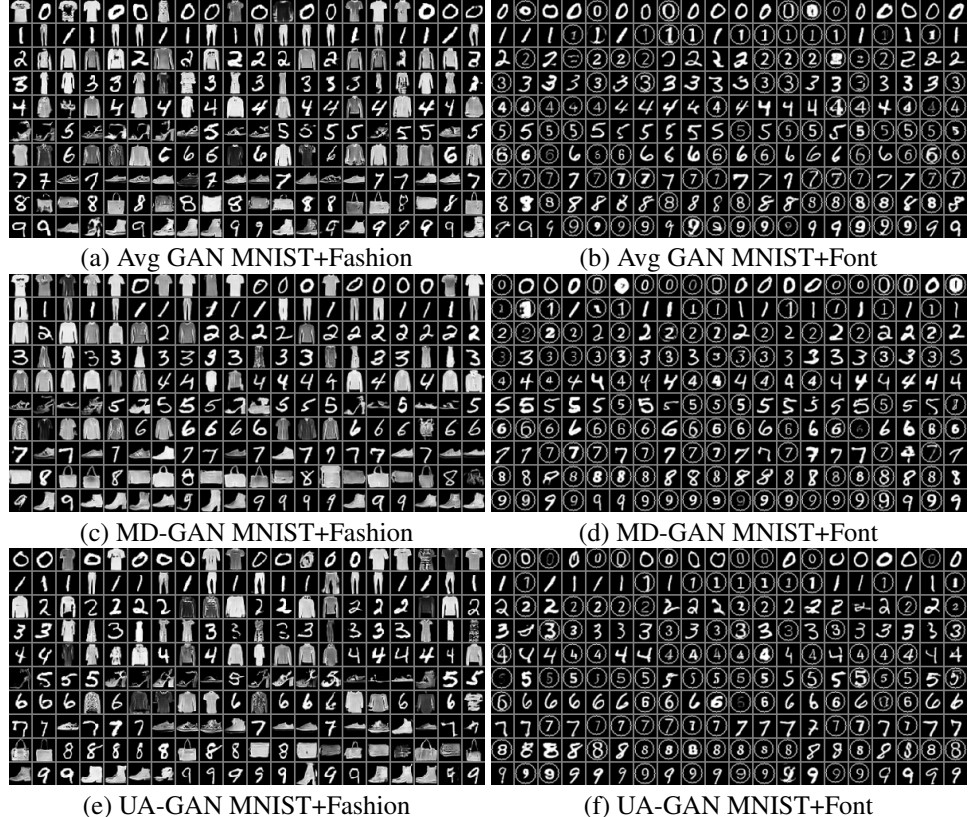

(a) Avg GAN MNIST+Fashion      (b) Avg GAN MNIST+Font

(c) MD-GAN MNIST+Fashion      (d) MD-GAN MNIST+Font

(e) UA-GAN MNIST+Fashion      (f) UA-GAN MNIST+Font

Figure 4: Synthetic images on the identical MNIST+Fashion dataset ((a),(c),(e)) and MNIST+Font dataset ((b),(d),(f)) using the average method, MD-GAN Hardy et al. (2019) and our UA-GAN method. All of the models could capture distributions over MNIST and Fashion/Font.

## B.2   ADDITIONAL EMPIRICAL RESULTS ON NON-IDENTICAL DISTRIBUTION

We provide additional synthetic images in non-identical distribution cases. See Fig. 5. By using average aggregation method, the synthetic image produced by Avg GAN and MD GAN only have Fashion images in (a), (c) and Font images in (b), (d). Our method in (e) and (f) could capture different patterns in MNIST + Fashion/Font and generate diverse images.

## B.3   EMPIRICAL RESULTS OF MIXING THREE DATASETS

We report the results of mixing the three datasets MNIST, FashionMNIST and Font. In the non-identical setting, we add MNIST data with a distinct class among 0~9. These data are distinguishable features for different sites. And we uniformly sample Fashion and Font data for all 10 distributed sites. These are common patterns across all sites. In the identical setting, all three datasets are uniformly distributed across the 10 sites. The quantitative results are shown in Table 3. The synthtic images are shown in Fig. 6 and Fig. 7. By using average aggregation method, the synthetic image produced by Avg-GAN and MD -GAN only have Fashion and Font images in Fig. 6(a), (b) . Our method in Fig. 6(c) could capture different patterns in MNIST + Fashion + Font and generate diverse images.

## B.4   EMPIRICAL RESULTS IN LARGER FEDERATED LEARNING SETTING

We report the results when using larger scale nodes($n = 50$) in distributed GAN methods(Avg-GAN, MD-GAN and UA-GAN). We uniformly split each individual site of the non-identical MNIST + Fashion dataset into 5 distributed sites. In total, we adopt 50 non-identical MNIST+Fashion datasets with 2380 MNIST and Fashion images each. The quantitative results are shown in Table 4, and the synthetic images are shown in Fig 8.

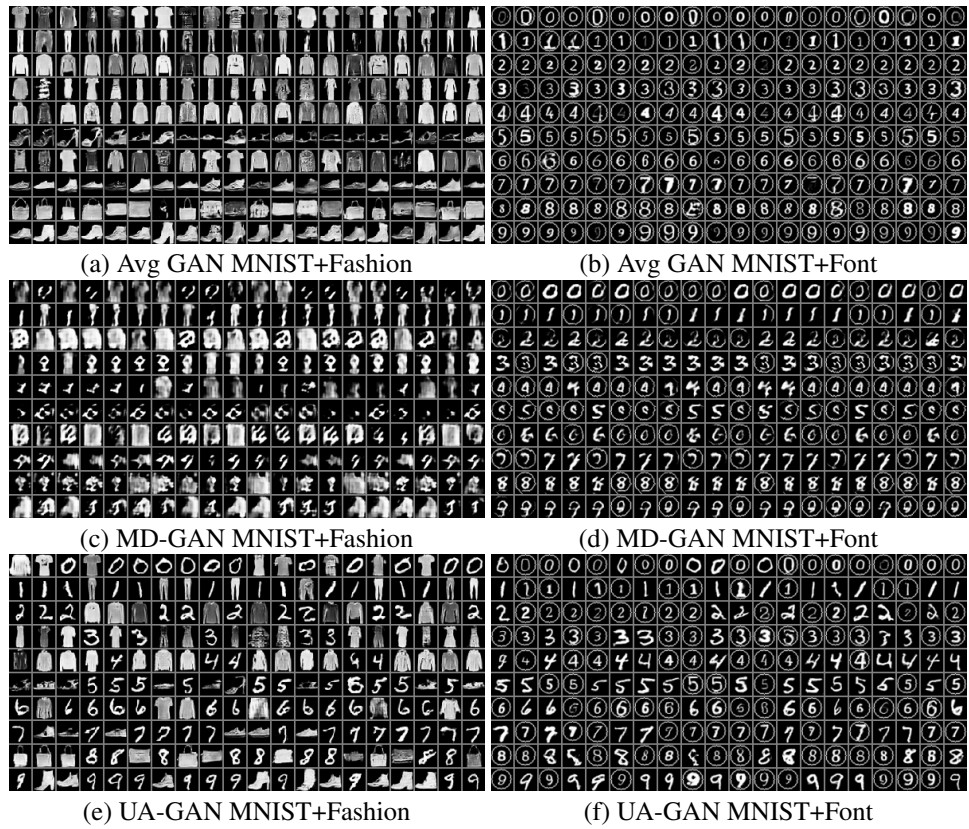

Figure 5: Additional synthetic images on the non-identical MNIST+Fashion dataset ((a),(c),(e)) and MNIST+Font dataset ((b),(d),(f)) using the average method, MD-GAN Hardy et al. (2019) and our UA-GAN method.

| Dataset | non-i.i.d | | | i.i.d | | |
|---|---|---|---|---|---|---|
| | Accuracy↑ | IS↑ | FID↓ | Accuracy↑ | IS↑ | FID↓ |
| Real | 0.955 | $3.426 \pm 0.023$ | 0 | 0.955 | $3.426 \pm 0.023$ | 0 |
| Centralized GAN | 0.943 | $3.031 \pm 0.016$ | 14.90 | 0.943 | $3.031 \pm 0.016$ | 14.90 |
| Avg GAN | 0.822 | $3.144 \pm 0.013$ | 41.63 | 0.936 | $2.877 \pm 0.013$ | 17.90 |
| MD-GAN | 0.567 | $3.035 \pm 0.011$ | 56.19 | 0.936 | $2.951 \pm 0.019$ | 16.81 |
| UA-GAN | 0.933 | $2.949 \pm 0.023$ | 20.80 | 0.923 | $2.875 \pm 0.013$ | 17.34 |

Table 3: The classification accuracy and IS, FID scores on mixture of three datasets.

| Dataset | non-i.i.d MNIST + Fashion (50 data sites) | | |
|---|---|---|---|
| | Accuracy↑ | IS↑ | FID↓ |
| Real | 0.943 | $3.620 \pm 0.021$ | 0 |
| Centralized GAN | 0.904 | $3.437 \pm 0.021$ | 8.35 |
| Avg GAN | 0.489 | $3.755 \pm 0.023$ | 90.36 |
| MD-GAN | 0.465 | $3.830 \pm 0.020$ | 89.36 |
| UA-GAN | 0.626 | $3.531 \pm 0.018$ | 53.26 |

Table 4: The classification accuracy and IS, FID scores on non-i.i.d mixture datasets for 50 distributed sites.

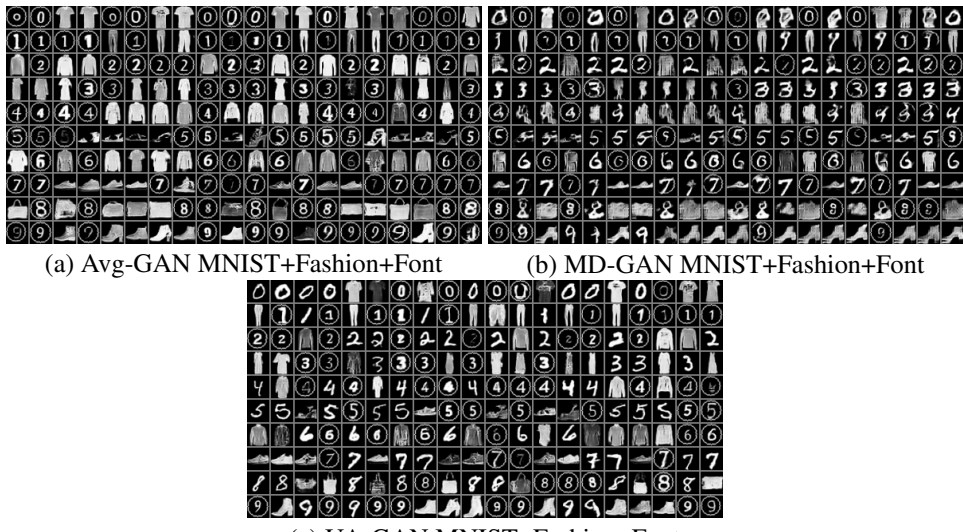

(a) Avg-GAN MNIST+Fashion+Font          (b) MD-GAN MNIST+Fashion+Font

(c) UA-GAN MNIST+Fashion+Font

Figure 6: Synthetic images on the non-identical MNIST+Fashion+Font dataset using the average method, MD-GAN Hardy et al. (2019) and our UA-GAN method.

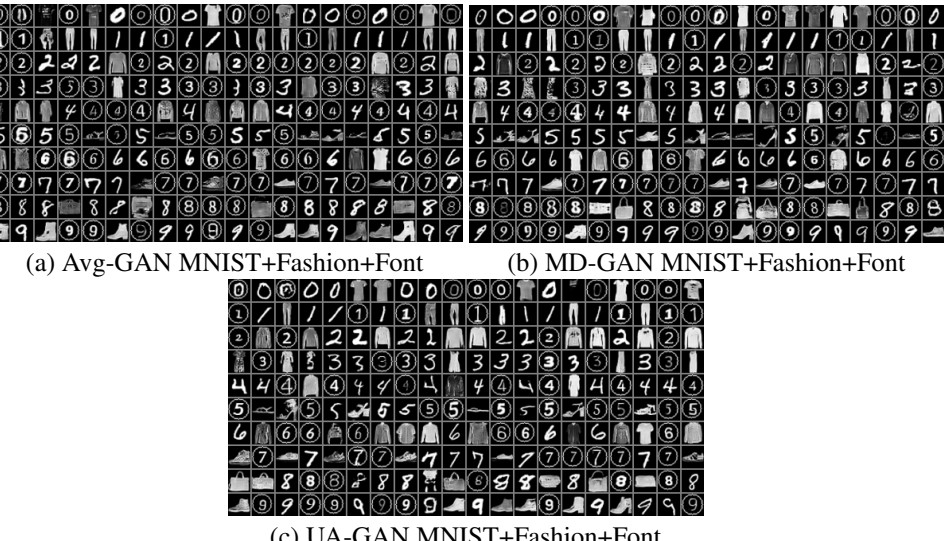

(a) Avg-GAN MNIST+Fashion+Font          (b) MD-GAN MNIST+Fashion+Font

(c) UA-GAN MNIST+Fashion+Font

Figure 7: Synthetic images on the identical MNIST+Fashion+Font dataset using the average method, MD-GAN Hardy et al. (2019) and our UA-GAN method.

## B.5 EMPIRICAL RESULTS IN UNCONDITIONAL SETTING

We report the results when using unconditional GAN in all methods (centralized, Avg-GAN, MD-GAN and UA-GAN). The quantitative results are shown in Table 5 and Table 6. The synthetic images are shown in Fig. 9 and Fig. 10. In the unconditional setting, the condition variable (labels) won't be given thus one can not directly apply the synthetic data in training classification model. Therefore we don't compute the classification accuracy in Table 5 and Table 6.

## B.6 EMPIRICAL RESULTS OF IMBALANCED DATASETS IN DIFFERENT SITES

We report the results when the sizes of the 10 sites are not the same. Based on the non-identical MNIST + fashionMNIST dataset, we reduce the sample sizes of the first 5 sites by half and keep the

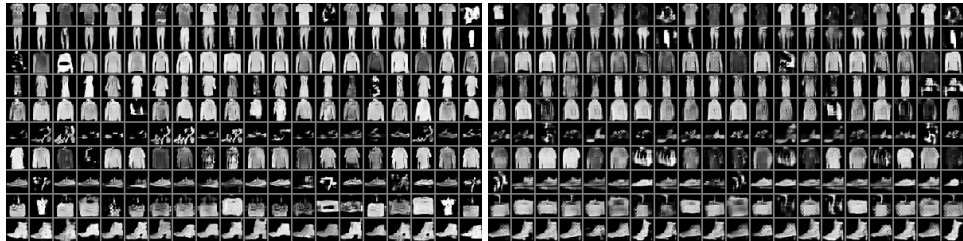

(a) Avg-GAN MNIST+Fashion(50 data sites)  (b) MD-GAN MNIST+Fashion(50 data sites)

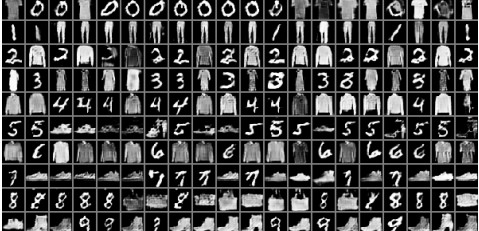

(c) UA-GAN MNIST+Fashion(50 data sites)

Figure 8: Synthetic images on the non-identical MNIST+Fashion dataset for 50 distributed sites using the average method, MD-GAN Hardy et al. (2019) and our UA-GAN method.

| Dataset | non-i.i.d MNIST + Fashion | | | non-i.i.d MNIST + Font | | |
|---|---|---|---|---|---|---|
| | Accuracy↑ | IS↑ | FID↓ | Accuracy↑ | IS↑ | FID↓ |
| Real | 0.943 | $3.620 \pm 0.021$ | 0 | 0.994 | $2.323 \pm 0.011$ | 0 |
| Centralized GAN | - | $3.387 \pm 0.019$ | 8.46 | - | $1.975 \pm 0.009$ | 17.56 |
| Avg GAN | - | $4.068 \pm 0.020$ | 61.56 | - | $1.547 \pm 0.005$ | 80.07 |
| MD-GAN | - | $2.852 \pm 0.021$ | 60.34 | - | $1.887 \pm 0.007$ | 36.36 |
| UA-GAN | - | $3.280 \pm 0.022$ | 22.34 | - | $1.985 \pm 0.013$ | 22.17 |

Table 5: The classification accuracy and IS, FID scores on two non-i.i.d mixture datasets in unconditional setting.

| Dataset | i.i.d MNIST + Fashion | | | i.i.d MNIST + Font | | |
|---|---|---|---|---|---|---|
| | Accuracy↑ | IS↑ | FID↓ | Accuracy↑ | IS↑ | FID↓ |
| Real | 0.943 | $3.620 \pm 0.021$ | 0 | 0.994 | $2.323 \pm 0.011$ | 0 |
| Centralized GAN | - | $3.387 \pm 0.019$ | 8.46 | - | $1.975 \pm 0.009$ | 17.56 |
| Avg GAN | - | $3.326 \pm 0.016$ | 9.68 | - | $1.918 \pm 0.006$ | 19.52 |
| MD-GAN | - | $3.428 \pm 0.025$ | 12.04 | - | $1.934 \pm 0.006$ | 18.85 |
| UA-GAN | - | $3.367 \pm 0.018$ | 9.99 | - | $1.937 \pm 0.009$ | 18.83 |

Table 6: The classification accuracy and IS, FID scores on two i.i.d mixture datasets in unconditional setting.

other 5 sites unchanged. In this case, the numbers of images in each site are shown in Table 7. The quantitative results are shown in Table 8. The synthetic images are shown in Fig. 11.

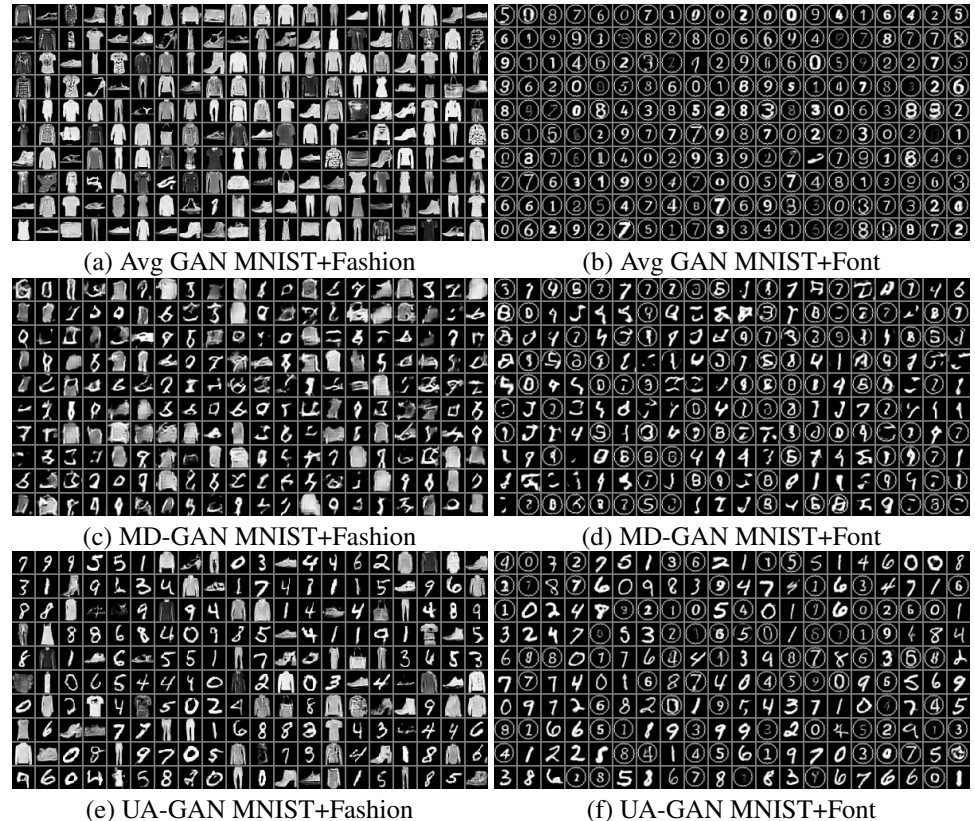

(a) Avg GAN MNIST+Fashion

(b) Avg GAN MNIST+Font

(c) MD-GAN MNIST+Fashion

(d) MD-GAN MNIST+Font

(e) UA-GAN MNIST+Fashion

(f) UA-GAN MNIST+Font

Figure 9: Synthetic images on the non-identical MNIST+Fashion dataset ((a),(c),(e)) and MNIST+Font dataset ((b),(d),(f)) in unconditional setting.

| | $D_0$ | $D_1$ | $D_2$ | $D_3$ | $D_4$ | $D_5$ | $D_6$ | $D_7$ | $D_8$ | $D_9$ |
|---|---|---|---|---|---|---|---|---|---|---|
| MNIST | 2917 | 3393 | 2944 | 3032 | 2939 | 5421 | 5918 | 6265 | 5851 | 5949 |
| Fashion | 3044 | 2978 | 3035 | 3033 | 2982 | 6000 | 6000 | 6000 | 6000 | 6000 |
| Total | 5961 | 6371 | 5979 | 6065 | 5921 | 11421 | 11918 | 12265 | 11851 | 11949 |

Table 7: The image numbers in each site in the imbalanced setting.

| | Accuracy↑ | IS↑ | FID↓ |
|---|---|---|---|
| Real | 0.939 | $3.580 \pm 0.039$ | 0 |
| Centralized GAN | 0.886 | $3.486 \pm 0.033$ | 10.87 |
| Avg GAN | 0.497 | $3.809 \pm 0.025$ | 74.45 |
| MD-GAN | 0.443 | $3.877 \pm 0.034$ | 85.61 |
| UA-GAN | 0.846 | $2.717 \pm 0.019$ | 30.30 |

Table 8: The classification accuracy and IS, FID scores on the imbalanced non-i.i.d mixture MNIST + fashionMNIST dataset.

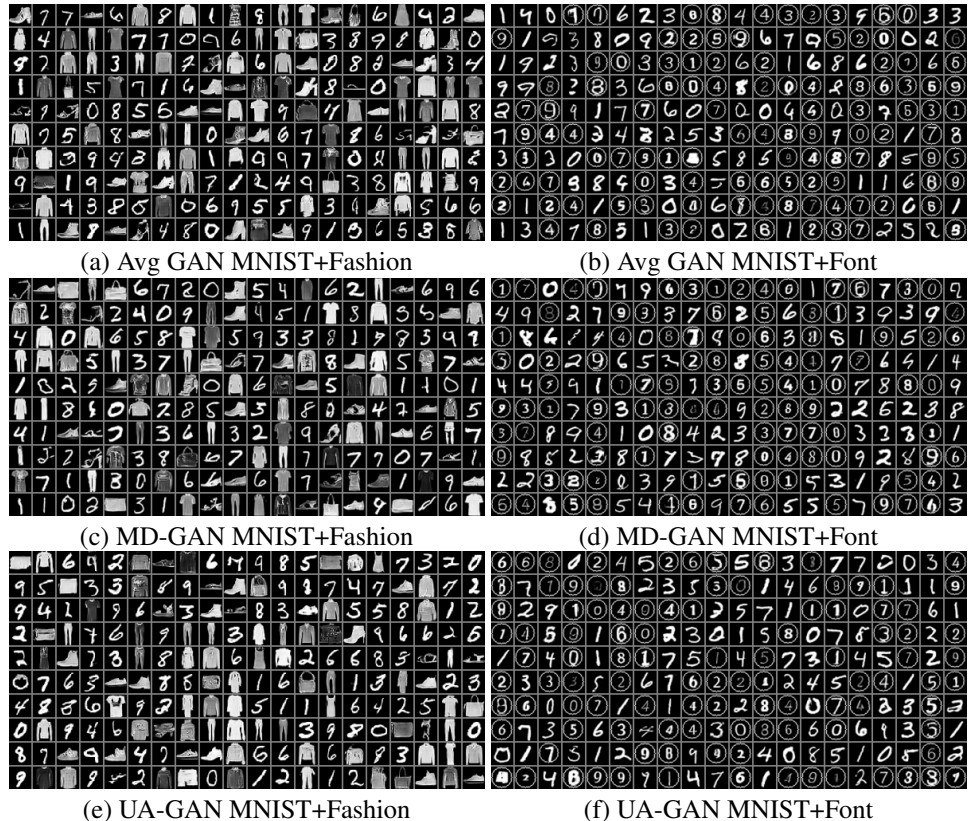

(a) Avg GAN MNIST+Fashion

(b) Avg GAN MNIST+Font

(c) MD-GAN MNIST+Fashion

(d) MD-GAN MNIST+Font

(e) UA-GAN MNIST+Fashion

(f) UA-GAN MNIST+Font

Figure 10: Synthetic images on the identical MNIST+Fashion dataset ((a),(c),(e)) and MNIST+Font dataset ((b),(d),(f)) in unconditional setting.

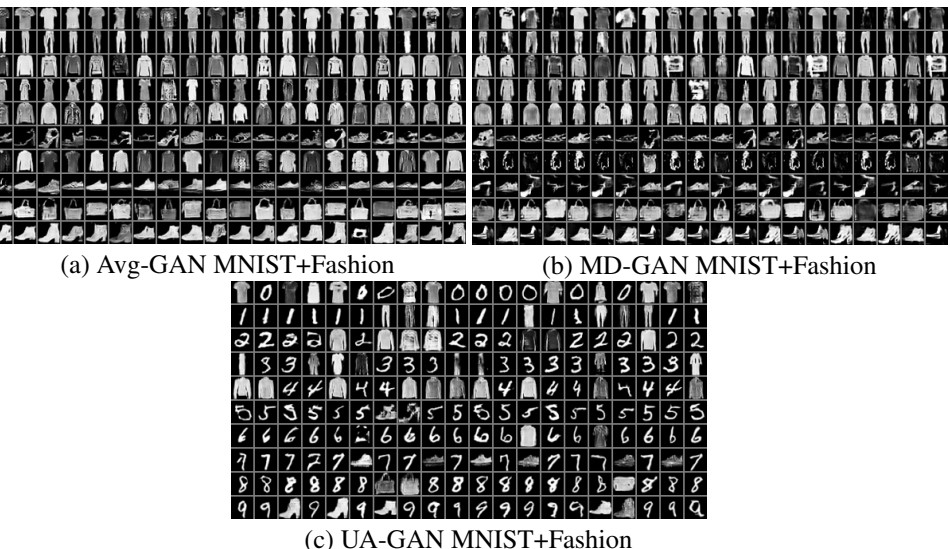

(a) Avg-GAN MNIST+Fashion

(b) MD-GAN MNIST+Fashion

(c) UA-GAN MNIST+Fashion

Figure 11: Synthetic images on the imbalanced non-identical MNIST+Fashion dataset.

