# OpenReview forum: "Training Federated GANs  with Theoretical Guarantees: A Universal Aggregation Approach"
_ICLR.cc/2021/Conference — Reject_

### Official Review · AnonReviewer3 · 2020-10-28
**Easy to read paper with interesting results**

**Rating:** 6
**Confidence:** 4

**Review:**

Summary:

The paper proposes a new method, UA-GAN to train GANs in a federated learning setup. The method simulates a central discriminator D_ua such that the odds values of the central discriminator is equivalent to the weighted sum of the local discriminators. The central generator is then trained based on the simulated central discriminator. The paper provides theoretical analysis on its proposed method and conducts experiments on toy datasets and mixtures of real world datasets to simulate a federated learning setup.

Pros:
- The problem of training GANs in a federated learning setup is an important problem to tackle as it naturally provides a secure way to train a generative model.
- The idea of simulated central discriminator based on the weighted odds value of the individual discriminators is quite interesting and seems to outperform a simple weighted sum of the discriminator gradients.
- The results provided in the experimental section are reasonable and outperforms similar GAN setups on non-identical data setups. I also appreciate the inclusion of an accuracy metric in evaluating the different methods as it shows the effectiveness of generated data in training of downstream tasks.

Cons:
- The paper provided a theoretical analysis of the performance of UA-GAN under conditions where some discriminators are trained sub-optimally. It would be nice to also have some experiments in the experimental section that shows the results of this setup i.e with smaller dataset size for some of the discriminators
- The experimental results will also be more convincing if results of training the UA-GAN on the mixture of all three real world datasets (Font, MNIST and Fashion-MNIST) are shown

Overall, the proposed method of training GANs in Federated Learning setup shows fairly convincing results. With some additional experimental results I believe this will be a good submission for ICLR.

==================

I thank the authors for the additional experiments which have marginally satisfied my initial concerns; ideally, more setup can be experimented. I keep my original rating.

---

> ### Author Response · Authors · 2020-11-18
> **Experiments for imbalanced dataset size setting and mix of MNIST, Fashion, Font  are added**
>
>  Thank you for your constructive and helpful comments. Here are our response to your suggestions.
>
> **Q1.** The paper provided a theoretical analysis of the performance of UA-GAN under conditions where some discriminators are trained sub-optimally. It would be nice to also have some experiments in the experimental section that shows the results of this setup i.e with smaller dataset size for some of the discriminators
>
> **Ans:**  Thank you for the suggestion. We ran additional experiments with imbalanced datasets. For the non-identical MNIST+ fashionMNIST setting, we reduced the sample sizes of the first 5 sites by half, so that their local discriminators cannot fully learn their local distributions. The other 5 sites remain unchanged. As shown in Appendix B.6 Table 8 and Fig. 11 of the revised version, UA-GAN still learned the mixture distribution to a satisfying level, whereas others failed.
>
>
> **Q2.** The experimental results will also be more convincing if results of training the UA-GAN on the mixture of all three real world datasets (Font, MNIST and Fashion-MNIST) are shown
>
> **Ans:** We performed experiments on the mixture of all three real world datasets and reported the detailed results in Appendix B.3 Table 3, Fig. 6 and Fig. 7 of the revised version. We observe a similar phenomenon as in the mixture-of-two datasets. In the non-iid setting, our UA-GAN is able to generate the mixture with both common patterns (Fashion and Font images) and site-specific images (different digits from MNIST) with high quality. Both baselines fail to do so. The Avg-GAN can only learn the common patterns, i.e., Fashion images and Font images. MD-GAN fails to produce high quality images.

---

### Official Review · AnonReviewer1 · 2020-10-29
**About the theory and experiments**

**Rating:** 5
**Confidence:** 4

**Review:**

This paper proposes federated GAN with multiple private discriminators. The authors also analyze the optimality of the proposed federated GAN. The review has several concerns on the current submission:
1.  Is minimax problem（5)  the total loss of UA-GAN that is optimized by Algorithm 1?  To solve the minimax problem (5),  D_{i} for i=1,2,..., K is coupled!.  Hence，why D_i defined in（2）is the solution of （5）when G is fixed？
2. When and how does the Nash equilibrium of minimax problem （5）holds？
3. Lack of experiments for federated unconditional GAN to verify the theory.

---

> ### Author Response · Authors · 2020-11-18
> **Questions are addressed and experiments for federated unconditional GAN are added.**
>
> Thank you for yourcomments. Here are our answers to your questions.
>
>
> **Q1**: Is minimax problem（5) the total loss of UA-GAN that is optimized by Algorithm 1? To solve the minimax problem (5), D_{i} for i=1,2,..., K is coupled!. Hence，why D_i defined in（2）is the solution of （5）when G is fixed？
>
> **Ans**: So far, there is not a unified minmax formulation for the federated GAN. Problem. Equation (5) is only the loss for the generator $G$. It assumes the local discriminators $D_j$’s have been learned optimally from the local distributions. All local discriminators are independently optimized. Each $D_j$ is learned by optimizing its own loss (Eq. (2)). They do not communicate with each other.
>
>
> **Q2**: When and how does the Nash equilibrium of minimax problem （5）holds？
>
> **Ans**: Since this is not a unified minmax problem, we do not pursue a Nash equilibrium. Theorem 1 proves the existence of a point at which the generator and all local discriminators achieve their own optimality, with respect to their own loss (Eq. (5) and (2)). Our analysis in Corollary 1 further shows that such balance is stable with regard to suboptimality.
>
> **Q3**: Lack of experiments for federated unconditional GAN to verify the theory.
>
> **Ans:** Thanks for the suggestion. We added experiments for federated unconditional GAN in Appendix Section B.5 of the revised version. The quantitative results for non-identical MNIST+fashionMNIST, MNIST+font and identical MNIST+fashionMNIST, MNIST+font are shown in Table 5 and Table 6, respectively. The corresponding image results are shown in Fig. 9 and Fig. 10. In the unconditional setting, we also observe significant superiority of our approach over other benchmarks.

---

### Official Review · AnonReviewer2 · 2020-10-30

**Rating:** 6
**Confidence:** 4

**Review:**

This paper proposes an algorithm for training GANs in federated learning setups. The federated learning task is to train a centralized generative model using the samples distributed across a group of local nodes in a network. Toward this goal, the paper proposes Universal Aggregation GAN (UA-GAN), in which at every iteration the server communicates a batch of samples produced by the generator to the local nodes and then the local nodes optimize their discriminator using their observed real and received fake samples. Overall, the UA-GAN approach is technically-sound and potentially useful for various distributed learning problems. However, I still have a few comments regarding the paper's theoretical and numerical results. I look forward to the authors' response to my comments.

My main concern is on training the discriminator in UA-GAN. According to Algorithm 1, UA-GAN trains K discriminator functions, i.e., one discrminator per local node in the network. In a typical federated learning problem, there might exist hundreds of nodes  where each node observes only tens or hundreds of training samples. In such scenarios, the generalization error of training one discriminator per local node can be very large, since every discriminator is trained using a limited number of real data points. On the other hand, the generalization error seems to be significantly smaller if one uses standard FedAvg for training a single discriminator network.  Currently, there are no theoretical guarantees or numerical analysis on the generalization properties of UA-GAN. I think analyzing the generalization behavior, either theoretically or numerically, is needed to address this comment, especially becuase the paper mostly focuses on the theoretical properties of UA-GAN.

The paper's numerical expeirments only consider setups with non-identical distributions across nodes. Also, in the experiments K is chosen to be moderately small while the number of training samples at every node is relatively large. I recommend providing some numerical results for larger federated learning settings with smaller training sets. For example, one can consider 100 local nodes with only 500 training samples available at every node. These numbers sound closer to a real federated learning problem than the ones used in the paper's experiments. In addition, lines 9 and 14 in Algorithm 1 should clearly state the optimization procedures for updating the discriminator and generator parameters instead of only referring to the optimization problems. In the current version, it remains unclear how many gradient steps are used to optimize the parameters at every iteration and how the performance of the trained GAN is affected by the number of gradient steps applied for optimizing the discriminator parameters at every communication round.

---

> ### Author Response · Authors · 2020-11-18
> **Questions about comparison with FedAvg are addressed. Experiments about small dataset are added. Detailed algorithm descriptions are added.**
>
> Thank you for your constructive feedback. Here are our answers to your questions.
>
> **Q1.**  When learning with many sites/nodes and fewer data per node, the generalization error of training one discriminator per local node can be very large. Why not using standard FedAvg to train a single centralized discriminator?
>
> **Ans:** The original motivation of federated GAN is for the cross-silo setting  (Kairouz et al. (2019) Table 1).  In such a setting, local sites/nodes represent institutions such as medical centers and education institutions. Each of these sites has a reasonable amount of data, but has strong privacy concerns. To extend to the setting with limited data per site, we could pre-train a discriminator using public datasets, and use this pre-trained model as the initial one of all local discriminators. It is possible to extend our theoretical framework to this solution in the future. We briefly mentioned this cross-silo assumption in Section 3. We will further clarify this in the future version.
>
> Under the cross-silo assumption (sufficient data per site), the whole idea of federated GAN is to circumvent passing the gradient information like FedAvg does. FedAvg suffers from privacy issues such as the gradient leakage (Zhu et al. 2019). In addition, its communication cost is huge especially for heavy models; the cost of a gradient is proportional to the model size. On the other hand, federated GAN only sends synthetic images and their local discriminator scores. This naturally alleviates privacy concerns. Furthermore, it saves a significant amount of communication cost. For example, a 128x128 size grey-scale synthetic image costs less than 1 mb, while the gradient of an image of the same size costs 40mb when the model is Resnet 18 .
>
>  **Q2.**  Analyze the generalization behavior, either theoretically or numerically. Provide some numerical results for larger federated learning settings with smaller training sets.
>
> **Ans:**  To the best of our knowledge, it remains an open problem to derive a non-pessimistic generalization gap for deep neural networks. Not mentioning a bound for GANs. However, your suggestion of numerical experiments is an excellent idea. We ran an additional experiment on MNIST+Fashion data. We allocated 50 sites, each with 2K images. Overall, we did observe a performance drop across all federated GAN methods. But UA-GAN still significantly outperforms other approaches. As we discussed before, this is out of the cross-silo setting and it can be improved by introducing pre-trained models using public datasets. Detailed results are provided in appendix section B.4 of the revised paper.
>
> **Q3.**  In addition, lines 9 and 14 in Algorithm 1 should clearly state the optimization procedures for updating the discriminator and generator parameters instead of only referring to the optimization problems.
>
> **Ans:** In each round we train the generator and discriminators for one iteration of gradient descent. We have added a precise description of the algorithm 1 in the appendix  section B named as Algorithm 2. We also add some details of applying the algorithm in the appendix

---

### Official Review · AnonReviewer4 · 2020-10-30
**An insufficiently clear paper**

**Rating:** 3
**Confidence:** 3

**Review:**




This paper studies gederated Generative Adversarial Networks (federated GANs).
In particular, the authors propose a new method, UA-GAN, which is claimed to be better than earlier approaches.

I have several concerns.  First, the writing can be improved significantly.  Second, the statements in the text are often rather vague.  This makes it hard to understand what are the results and to verify their correctness.   E.g., theorems are formulated either vaguely or unprecisely.

Moreover, the theorems refer to "optimal discriminators", however there is no guarantee that one can find such an optimal discriminator or that one can even verify whether a discriminator is optimal.  (In fact, the paper doesn't define the term "optimal" precisely)   Even if it would be possible to learn an optimal discriminator, then there is no bound on the amount of time this would take (as that depends on various parameters of the learning problem and the learning algorithm).

The paper contains more vague statements, e.g., Remark 1 claims the method preserves privacy, but doesn't define what information is kept private.   For example, it seems the size of the private datasets is needed for the central weighting and hence made public.  Moreover, there is no proof that from the information leaving the individual centers nothing about the sensitive data can be inferred.  For example, from the point of view of differential privacy, even if only aggregates are revealed, if their revealed value is exact then probably (epsilon,delta) - differential privacy is not guaranteed.

In conclusion, it is hard for the average reader to understand the paper due to a lack of precision, and the paper insufficiently specifies definitions, assumptions made, and precise formulations of results.



Some details:

* The first line of the abstract suggests that GANs are federated networks, while the second line of the abstract correctly states that this is called "federated GAN" and the first line of the introduction correctly describes GAN as generating realistic data.
* Algorithm 1: "Eq. equation 2." -> repetition
* last line page 3: "algorithm equation 1" -> "Algorithm 1"
* top of page 4: the generator G(z) seems to depend on an argument z of which the nature is not revealed immediately.  Later, Equation (2) suggests that z can be drawn from \mathcal{N}(0,1) and hence is a real number.  It is unclear why the argument of G would be just one-dimensional.
* top of page 4: while the word "discriminator" is used very frequently, no precise definition of this concept is provided, nor an explanation of how the discriminators are obtained.  With the help of Eq (2) some readers may be able to guess that D_j must have values in the open interval (0,1).
* Definition 1: as the symbol q is already used for distributions, it is preferably to not use it for "a probability in (0,1)" too.
* After definition 1: "The central idea of UA framework" -> "framework" needs an article
* just before Section 3.1:  We will provide error bound -> ... an error bound
* Theorem 1 uses "Jenson Shannon divergence loss", which isn't defined in the paper (and no definition is cited).  "Jenson Shannon divergence" is a somewhat well-known concept in probability theory, but even for those knowing this it is unclear how to get from it to "Jenson Shannon divergence loss".
* Equation (6) in Theorem 1 seems to give an expression similar to the Jenson-Shannon divergence definition, but doesn't appear a statement the theorem is claiming to be true.  The rest of the sentence refers to q^* and q, but q is a bound variable in Eq (6), i.e., it has no meaning outside the scope of "argmin_q", and q^* doesn't occur in the formula (so why do you say "where q^* = ..." ?).  It is hence hard to parse the theorem statement and discover what is the claim exactly.
* A proof is provided in appendix.  However, the proof first says "To prove the theorem, we first introduce the following Lemma", the text next states lemma 1, but never returns to the proof of Theorem 1 (as the next title says "Proof of Theorem 4").

---

> ### Author Response · Authors · 2020-11-18
> **We thank the reviewer for pointing out various presentation issues. We have improved the manuscript accordingly. Below we address specific concerns.**
>
> **Q1:** The optimal discriminators are not guaranteed to be found.
>
> **Ans**: Our theorem 1 proves that there exists a point at which the optimal generator and optimal local discriminators coexist. This is following the existing theoretical framework for standard GANs (Goodfellow et al. 14). Proving whether the optimality can be achieved is very challenging. We are not aware of any such theoretical results even in the standard GAN literature. In fact, our theorem 2 and corollary 1 are the only results toward this direction. We showed that even if the discriminator is slightly suboptimal, the generator optimized over the Jenson Shannon divergence loss will deteriorate, but in a controlled manner. These results also apply to a standard GAN.
>
> **Q2:** The paper contains more vague statements, e.g., Remark 1 claims the method preserves privacy, but doesn't define what information is kept private. For example, it seems the size of the private datasets is needed for the central weighting and hence made public. Moreover, there is no proof that from the information leaving the individual centers nothing about the sensitive data can be inferred. For example, from the point of view of differential privacy, even if only aggregates are revealed, if their revealed value is exact then probably (epsilon,delta) - differential privacy is not guaranteed.
>
> **Ans:** Federated GAN largely mitigates the risk of leaking privacy as it only passes around synthetic images and their discriminator scores. It is true that the local dataset size is released. But it is unlikely this would lead to a significant damage. It is interesting and possible to incorporate a differential privacy mechanism into federated GANs. This is a future direction worth pursuing.

---

> > ### Author Response · Authors · 2020-11-18
> > **We've fixed typos in the paper and clarify your questions in the revision.**
> >
> > **Q3:** The first line of the abstract suggests that GANs are federated networks, while the second line of the abstract correctly states that this is called "federated GAN" and the first line of the introduction correctly describes GAN as generating realistic data.
> >
> > **Ans:** We are having trouble understanding this comment/suggestion. Could you please elaborate? In general, GAN is a method to learn to generate realistic data. When incorporated into the federated learning setting, we call it a federated GAN. But the implementation of GAN in the federated setting is nontrivial, as we showed in the paper.
> >
> > **Q4:** top of page 4: the generator G(z) seems to depend on an argument z of which the nature is not revealed immediately.
> >
> > **Ans:** z is a multivariate random noise input to the generator $G$. This is a very standard notation. It was first defined in the seminal work (Goodfellow et al., 2014) and has been widely adapted in the GAN literature.
> >
> > **Q5:** Equation (2) suggests that z can be drawn from \mathcal{N}(0,1) and hence is a real number. It is unclear why the argument of G would be just one-dimensional.
> >
> > **Ans:** Thanks for pointing it out. This is a typo. z is a $d$-dimensional vector sampled from a multivariate Gaussian, $\mathcal{N}(0, I_d)$. We fixed this in Eq. (2) of the updated version.
> >
> > **Q5:** top of page 4: while the word "discriminator" is used very frequently, no precise definition of this concept is provided, nor an explanation of how the discriminators are obtained. With the help of Eq (2) some readers may be able to guess that D_j must have values in the open interval (0,1).
> >
> > **Ans:** The discriminator learns the posterior probability of an image being real. Therefore its value is between 0 and 1. It is a very standard term in the GAN literature, first defined in (Goodfellow et al. 2014).
> >
> > **Q6:** Definition 1: as the symbol q is already used for distributions, it is preferably to not use it for "a probability in (0,1)" too.
> >
> > **Ans:** we have changed $q$ to $\phi$ in the updated version.
> >
> > **Q7:** Theorem 1 uses "Jenson Shannon divergence loss", which isn't defined in the paper (and no definition is cited). "Jenson Shannon divergence" is a somewhat well-known concept in probability theory, but even for those knowing this it is unclear how to get from it to "Jenson Shannon divergence loss".
> >
> > **Ans:** Thanks for pointing this out. Assuming optimal discriminators, Eq. (6) is the Jenson Shannon divergence loss. In the standard GAN setting, Jensen Shannon divergence loss, first defined in (Goodfellow et al., 2014), is the generator’s loss when the discriminator is optimal. We will clarify and add citations.
> >
> > **Q8:** In Theorem 1 the sentence below Equation (6) refers to q^* and q, but q is a bound variable in Eq (6), i.e., it has no meaning outside the scope of "argmin_q", and q^* doesn't occur in the formula (so why do you say "where q^* = ..." ?). It is hence hard to parse the theorem statement and discover what is the claim exactly.
> >
> > **Ans:** $q^\ast$ is the optimal $q$ in Equation (6). The theorem states that $q^\ast$ is the true distribution $p$. We have revised the theorem statement.
> >
> >
> > **Q9:** A proof is provided in appendix. However, the proof first says "To prove the theorem, we first introduce the following Lemma", the text next states lemma 1, but never returns to the proof of Theorem 1 (as the next title says "Proof of Theorem 4").
> >
> > **Ans:** For convenience, in the appendix, we restated Theorem 1 and called it Theorem 4. The proof of theorem 4 is the proof of theorem 1. We have cleared this in the revised version.

---

### Decision · Program_Chairs · 2021-01-07
**Final Decision**

**Decision:**

Reject

**Comment:**

The paper presents a provable correct framework, namely Universal Aggregation, for training GANs in federated learning scenarios. It aims to address an important problem. The proposed solution is well-grounded with theoretical analysis and promising empirical results.

The paper receives mixed ratings and therefore there were extensive discussions. One the positive end, some reviewers think that the authors' feedback provide clarification to confusing part of the paper; on the negative side, the authors feedback also confirms some of the concerns raised in the reviews:

1.  It was confirmed that there is no guarantee that one can find an (nearly) optimal discriminator, which decreases the impact of the work, as in practice we work with non-optimal discriminators and hence some of the results couldn't apply.

2. It was confirmed that no privacy guarantees can be given. This is concerning since the complexity of GANs won't prohibit skilled attackers from inferring some information.

While it is true that some of the guarantees would be hard to achieve even for a traditional GAN, the paper sets up a high-expectation at the beginning of the paper,  but fails to satisfy the readers with enough evidence.

In addition, the writing can be significantly improved to ensure precise formulations and consistency; the added experiment results are useful, but stronger empirical results could help alleviate the issues in theoretical results.

In summary, the paper has built solid foundations for a good piece of work, but the current version could benefit from one more round of revision to become a strong publication in the future.